# BLACK-BOX ADVERSARIAL ATTACKS WITH BAYESIAN OPTIMIZATION

## ABSTRACT

We focus on the problem of black-box adversarial attacks, where the aim is to generate adversarial examples using information limited to loss function evaluations of input-output pairs. We use Bayesian optimization (BO) to specifically cater to scenarios involving low query budgets to develop query efficient adversarial attacks. We alleviate the issues surrounding BO in regards to optimizing high dimensional deep learning models by effective dimension upsampling techniques. Our proposed approach achieves performance comparable to the state of the art black-box adversarial attacks albeit with a much lower average query count. In particular, in low query budget regimes, our proposed method reduces the query count up to $80\%$ with respect to the state of the art methods.

## 1 INTRODUCTION

Neural networks are now well-known to be vulnerable to *adversarial examples*: additive perturbations that, when applied to the input, change the network's output classification (Goodfellow et al., 2014). Work investigating this lack of robustness to adversarial examples often takes the form of a back-and-forth between newly proposed *adversarial attacks*, methods for quickly and efficiently crafting adversarial examples, and corresponding defenses that modify the classifier at either training or test time to improve robustness. The most successful adversarial attacks use gradient-based optimization methods (Goodfellow et al., 2014; Madry et al., 2017), which require complete knowledge of the architecture and parameters of the target network; this assumption is referred to as the *white-box* attack setting. Conversely, the more realistic *black-box* setting requires an attacker to find an adversarial perturbation without such knowledge: information about the network can be obtained only through querying the target network, i.e., supplying an input to the network and receiving the corresponding output.

In real-world scenarios, it is extremely improbable for an attacker to have unlimited bandwidth to query a target classifier. In evaluation of black box attacks, this constraint is usually formalized via the introduction of a *query budget*: a maximum number of queries allowed to the model per input, after which an attack is considered to be unsuccessful. Several recent papers have proposed attacks specifically to operate in this query-limited context (Ilyas et al., 2019; 2018; Chen et al., 2017; Tu et al., 2019; Moon et al., 2019); nevertheless, these papers typically consider query budgets on the order of 10,000 or 100,000. This leaves open questions as to whether black-box attacks can successfully attack a deep network based classifier in severely query limited settings, e.g., with a query budget of 100-200. In such a query limited regime, it is natural for an attacker to use the entire query budget, so we ask the pertinent question: *In a constrained query limited setting, can one design query efficient yet successful black box adversarial attacks?*

This work proposes a black-box attack method grounded in Bayesian optimization (Jones et al., 1998; Frazier, 2018), which has recently emerged as a state of the art black-box optimization technique in settings where minimizing the number of queries is of paramount importance. Straight-forward application of Bayesian optimization to the problem of finding adversarial examples is not feasible: the input dimension of even a small neural network-based image classifier is orders of magnitude larger than the standard use case for Bayesian optimization. Rather, we show that we can bridge this gap by performing Bayesian optimization in a reduced-dimension setting and upsampling to obtain our final perturbation. We explore several upsampling techniques and find that a relatively simple nearest-neighbor upsampling method allows us to sufficiently reduce the optimization prob-

lem dimension such that Bayesian optimization can find adversarial perturbations with more success than existing black-box attacks in query-constrained settings.

We compare the efficacy of our adversarial attack with a set of experiments attacking three of the most commonly used pretrained ImageNet (Deng et al., 2009) classifiers: ResNet50 (He et al., 2015), Inception-v3 (Szegedy et al., 2015), and VGG16-bn (Simonyan & Zisserman, 2014). Results from these experiments show that with very small query budgets (under 200 queries), the proposed method BAYES-ATTACK achieves success rates comparable to or exceeding existing methods, and does so with far smaller average and median query counts. Further experiments are performed on the MNIST dataset to compare how various upsampling techniques affect the attack accuracy of our method. Given these results we argue that, despite being a simple approach (indeed, largely *because* it is such a simple and standard approach for black-box optimization), Bayesian Optimization should be a standard baseline for any black-box adversarial attack task in the future, especially in the small query budget regime.

## 2 RELATED WORK

Within the black-box setting, adversarial attacks can be further categorized by the exact nature of the information received from a query. The most closely related work to our approach are *score-based* attacks, where queries to the network return the entire output layer of the network, either as logits or probabilities. Within this category, existing approaches draw from a variety of optimization fields and techniques. One popular approach in this area is to attack with zeroth-order methods via some method of derivative-free gradient estimation, as in methods proposed in Ilyas et al. (2019), which uses time-dependent and data-dependent priors to improve the estimate, as well as Ilyas et al. (2018), which replaces the gradient direction found using natural evolution strategies (NES). Other methods search for the best perturbation outside of this paradigm; Moon et al. (2019) cast the problem of finding an adversarial perturbation as a discrete optimization problem and use local search methods to solve. These works all search for adversarial perturbations within a search space with a hard constraint on perturbation size; other work (Chen et al., 2017; Tu et al., 2019) incorporates a soft version of this constraint and performs coordinate descent to decrease the perturbation size while keeping the perturbed image misclassified. The latter of these methods incorporates an autoencoder-based upsampling method with which we compare in Section 5.3.1.

One may instead assume that only part of the information from the network's output layer is received as the result of a query. This can take the form of only receiving the output of the top $k$ predicted classes (Ilyas et al., 2018), but more often the restrictive *decision-based* setting is considered. Here, queries yield only the predicted class, with no probability information. The most successful work in this area is in Cheng et al. (2018), which reformulates the problem as a search for the direction of the nearest decision boundary and solves using a random gradient-free method, and in Brendel et al. (2017) and Chen et al. (2019), both of which use random walks along the decision boundary to perform an attack. The latter work significantly improves over the former with respect to query efficiency, but the number of queries required to produce adversarial examples with small perturbations in this setting remains in the tens of thousands.

A separate class of *transfer-based* attacks train a second, fully-observable substitute network, attack this network with white-box methods, and transfer these attacks to the original target network. These may fall into one of the preceding categories or exist outside of the distinction: in Papernot et al. (2016), the substitute model is built with score-based queries to the target network, whereas Liu et al. (2016) trains an ensemble of models without directly querying the network at all. These methods come with their own drawbacks: they require training a substitute model, which may be costly or time-consuming, and overall attack success tends to be lower than that of gradient-based methods.

Finally, there has been some recent interest in leveraging Bayesian optimization for constructing adversarial perturbations. Bayesian optimization (BO) has played a supporting role in several methods. For example, Zhao et al. (2019) use BO to solve the $\delta$-step of an alternating direction of method multipliers (ADMM) approach, Co et al. (2018) search within a set of procedural noise perturbations using BO and Gopakumar et al. (2018) use BO to find maximal distortion error by optimizing perturbations defined using 3 parameters. On the other hand, prior work in which Bayesian optimization plays a central role performs experiments only in relatively low-dimensional problems, highlighting the main challenge of its application: Suya et al. (2017) examines an attack on a spam

email classifier with 57 input features, and in Co (2017) image classifiers are attacked but notably do not scale beyond MNIST classifiers. In contrast to these past works, the main contribution of this paper is to show that Bayesian Optimization presents a *scalable, query-efficient* approach for large-scale black-box adversarial attacks, when combined with upsampling procedures.

## 3 PROBLEM FORMULATION

The following notation and definitions will be used throughout the remainder of the paper. Let $F$ be the target neural network. We assume that $F : \mathbb{R}^d \to [0,1]^K$ is a $K$-class image classifier that takes normalized inputs: each dimension of an input $\mathbf{x} \in \mathbb{R}^d$ represents a single pixel and is bounded between 0 and 1, $y \in \{1, \cdots K\}$ denotes the original label, and the corresponding output $F(\mathbf{x})$ is a $K$-dimensional vector representing a probability distribution over classes.

Rigorous evaluation of an adversarial attack requires careful definition of a *threat model*: a set of formal assumptions about the goals, knowledge, and capabilities of an attacker (Carlini & Wagner, 2017). We assume that, given a correctly classified input image $\mathbf{x}$, the goal of the attacker is to find a perturbation $\boldsymbol{\delta}$ such that $\mathbf{x} + \boldsymbol{\delta}$ is misclassified, i.e., $\arg\max_k F(\mathbf{x} + \boldsymbol{\delta}) \neq \arg\max_k F(\mathbf{x})$. We operate in the score-based black-box setting, where we have no knowledge of the internal workings of the network, and a query to the network $F$ yields the entire corresponding $K$-dimensional output. To enforce the notion that the adversarial perturbation should be small, we take the common approach of requiring that $\|\boldsymbol{\delta}\|_p$ be smaller than a given threshold $\epsilon$ in some $\ell_p$ norm, where $\epsilon$ varies depending on the classifier. This work considers the $\ell_\infty$ norm, but our attack can easily be adapted to other norms. Finally, we denote the query budget with $t$; if an adversarial example is not found after $t$ queries to the target network, the attack fails.

As in most work, we pose the attack as a constrained optimization problem. We use an objective function suggested by Carlini & Wagner (2017) and used in Tu et al. (2019); Chen et al. (2017):

$$\max_{\boldsymbol{\delta}} f(\mathbf{x}, y, \boldsymbol{\delta}) \quad \text{subject to} \ \ \|\boldsymbol{\delta}\|_p \leq \epsilon \ \text{ and } \ (\mathbf{x} + \boldsymbol{\delta}) \in [0,1]^d, \tag{1}$$

$$\text{where} \quad f(\mathbf{x}, y, \boldsymbol{\delta}) = \big\{ \max_{k \neq y} \log[F(\mathbf{x} + \boldsymbol{\delta})]_k - \log[F(\mathbf{x} + \boldsymbol{\delta})]_y \big\}.$$

Most importantly, the input $\mathbf{x} + \boldsymbol{\delta}$ to $f$ is an adversarial example for $F$ if and only if $f(\mathbf{x}, y, \boldsymbol{\delta}) > 0$.

We briefly note that the above threat model and objective function were chosen for simplicity and for ease of directly comparing with other black box attacks, but the attack method we propose is compatible with many other threat models. For example, we may change the goals of the attacker or measure $\delta$ in $\ell_1$ or $\ell_2$ norms instead of $\ell_\infty$ with appropriate modifications to the objective function and constraints in equation 1.

## 4 MODEL FRAMEWORK

In this section, we present the proposed black-box attack method. We begin with a brief description of Bayesian optimization (Jones et al., 1998) followed by its application to generate black-box adversarial examples. Finally, we describe our method for attacking a classifier trained with high-dimensional inputs (e.g. ImageNet) in a query-efficient manner.

### 4.1 BAYESIAN OPTIMIZATION

Bayesian Optimization (BO) is a method for black box optimization particularly suited to problems with low dimension and expensive queries. Bayesian Optimization consists of two main components: a Bayesian statistical model and an acquisition function. The Bayesian statistical model, also referred to as the surrogate model, is used for approximating the objective function: it provides a Bayesian posterior probability distribution that describes potential values for the objective function at any candidate point. This posterior distribution is updated each time we query the objective function at a new point. The most common surrogate model for Bayesian optimization are Gaussian processes (GPs) (Rasmussen & Williams, 2005), which define a prior over functions that are cheap to evaluate and are updated as and when new information from queries becomes available. We model the objective function $h$ using a GP with prior distribution $\mathcal{N}(\mu_0, \Sigma_0)$ with constant mean

function $\mu_0$ and Matern kernel (Shahriari et al., 2016; Snoek et al., 2012) as the covariance function $\Sigma_0$, which is defined as:

$$\Sigma_0(\mathbf{x}, \mathbf{x}') = \theta_0^2 \exp(-\sqrt{5}r)\left(1 + \sqrt{5}r + \frac{5}{3}r^2\right),$$

$$r^2 = \sum_{i=1}^{d'} \frac{(x_i - x_i')^2}{\theta_i^2}$$

where $d'$ is the dimension of input and $\{\theta_i\}_{i=0}^{d'}$ and $\mu_0$ are hyperparameters. We select hyperparameters that maximize the posterior of the observations under a prior (Shahriari et al., 2016; Frazier, 2018).

The second component, the acquisition function $\mathcal{A}$, assigns a value to each point that represents the utility of querying the model at this point given the surrogate model. We sample the objective function $h$ at $\mathbf{x}_n = \arg\max_{\mathbf{x}} \mathcal{A}(\mathbf{x}|\mathcal{D}_{1:n-1})$ where $\mathcal{D}_{1:n-1}$ comprises of $n-1$ samples drawn from $h$ so far. Although this itself may be a hard (non-convex) optimization problem to solve, in practice we use a standard approach and approximately optimize this objective using the LBFGS algorithm. There are several popular choices of acquisition function; we use expected improvement (EI) (Jones et al., 1998), which is defined as

$$\text{EI}_n(\mathbf{x}) = \mathbb{E}_n\left[\max\ (h(\mathbf{x}) - h_n^*, 0)\right], \tag{2}$$

where $\mathbb{E}_n[\cdot] = \mathbb{E}[\cdot|\mathcal{D}_{1:n-1}]$ denotes the expectation taken over the posterior distribution given evaluations of $h$ at $\mathbf{x}_1, \cdots, \mathbf{x}_{n-1}$, and $h_n^*$ is the best value observed so far.

Bayesian optimization framework as shown in Algorithm 2 runs these two steps iteratively for the given budget of function evaluations. It updates the posterior probability distribution on the objective function using all the available data. Then, it finds the next sampling point by optimizing the acquisition function over the current posterior distribution of GP. The objective function $h$ is evaluated at this chosen point and the whole process repeats.

In theory, we may apply Bayesian optimization directly to the optimization problem in equation 1 to obtain an adversarial example, stopping once we find a point where the the objective function rises above $0$. In practice, Bayesian optimization's speed and overall performance fall dramatically as the input dimension of the problem increases. This makes running Bayesian optimization over high dimensional inputs such as ImageNet (input dimension $3 \times 299 \times 299 = 268203$) practically infeasible; we therefore require a method for reducing the dimension of this optimization problem.

### 4.2 BAYES-ATTACK: GENERATING ADVERSARIAL EXAMPLES USING BAYESIAN OPTIMIZATION

Images tend to exhibit spatial local similarity i.e. pixels that are close to each other tend to be similar. Ilyas et al. (2019) showed that this similarity also extends to gradients and used this to reduce query complexity. Our method uses this data dependent prior to reduce the search dimension of the perturbation. We show that the adversarial perturbations also exhibit spatial local similarity and we do not need to learn the adversarial perturbation conforming to the actual dimensions of the image. Instead, we learn the perturbation in a much lower dimension. We obtain our final adversarial perturbation by interpolating the learned, low-dimension perturbation to the original input dimension.

We define the objective function for running the Bayesian optimization in low dimension in Algorithm 1. We let $\Pi_{B(\mathbf{0},\epsilon)}^p$ be the projection onto the $\ell_\infty$ ball of radius $\epsilon$ centered at origin. Our method finds a low dimension perturbation and upsamples to obtain the adversarial perturbation. Since this upsampled image may not lie inside the ball of radius $\epsilon$ centered at the origin, we project back to ensure $\|\delta\|_\infty$ remains bounded by $\epsilon$. With the perturbation $\delta$ in hand, we compute the objective function of the original optimization problem defined in equation 1.

We describe the complete algorithm our complete framework in Algorithm 2 where $\mathbf{x}_0 \in \mathbb{R}^d$ and $y_0 \in \{1, \ldots, K\}$ denote the original input image and label respectively. The goal is to learn an adversarial perturbation $\boldsymbol{\delta} \in \mathbb{R}^{d'}$ in much lower dimension, i.e., $d' << d$. We begin with a small dataset $\mathcal{D} = \{(\boldsymbol{\delta}_1, v_1), \cdots, (\boldsymbol{\delta}_{n_0}, v_{n_0})\}$ where each $\boldsymbol{\delta}_n$ is a $d'$ dimensional vector sampled from a

---

**Algorithm 1** Objective Function

---

1: **procedure** OBJ-FUNC($\mathbf{x}_0, y_0, \boldsymbol{\delta}$)
2:     // $\epsilon$ is the given perturbation
3:     $\boldsymbol{\delta}' \leftarrow \text{Upsample}(\boldsymbol{\delta})$         ▷ Upsampling low dimension perturbation to input dimension
4:     $\boldsymbol{\delta}' \leftarrow \Pi^p_{B(\mathbf{0},\epsilon)}(\boldsymbol{\delta}')$         ▷ Projecting perturbation on $\ell_p$-ball around $x_0$
5:     $v \leftarrow f(\mathbf{x}_0, y_0, \boldsymbol{\delta}')$         ▷ Quering the model
6:     **return** $v$

---

**Algorithm 2** Adversarial Attack using Bayesian Optimization

---

1: **procedure** BAYES-ATTACK($x_0, y_0$)
2:     $\mathcal{D} = \{(\boldsymbol{\delta}_1, v_1), \cdots, (\boldsymbol{\delta}_{n_0}, v_{n_0})\}$         ▷ Quering randomly chosen $n_0$ points.
3:     Update the GP on $\mathcal{D}$         ▷ Updating posterior distribution using available points
4:     $t \leftarrow n_0$         ▷ Updating number of queries till now
5:     **while** $t \leq T$ **do**
6:         $\boldsymbol{\delta}_t \leftarrow \arg\max_{\boldsymbol{\delta}} \mathcal{A}(\boldsymbol{\delta} \mid \mathcal{D})$         ▷ Optimizing the acquisition function over the GP
7:         $v_t \leftarrow \text{OBJ-FUNC}(\mathbf{x}_0, y_0, \boldsymbol{\delta})$         ▷ Querying the model
8:         $t \leftarrow t + 1$
9:         **if** $v_t \leq 0$ **then**
10:             $\mathcal{D} \leftarrow \mathcal{D} \cup (\boldsymbol{\delta}_t, v_t)$ and update the GP         ▷ Updating posterior distribution
11:         **else**
12:             **return** $\boldsymbol{\delta}_t$         ▷ Adversarial attack successful
13:     **return** $\boldsymbol{\delta}_t$         ▷ Adversarial attack unsuccessful

---

given distribution and $v_n$ is the function evaluation at $\boldsymbol{\delta}_n$ i.e $v_n = \text{OBJ-FUNC}(\mathbf{x}_0, y_0, \boldsymbol{\delta}_n)$. We iteratively update the posterior distribution of the GP using all available data and query new perturbations obtained by maximizing the acquisition function over the current posterior distribution of GP until we find an adversarial perturbation or run out of query budget. The Bayesian optimization iterations run in low dimension $d'$ but for querying the model we upsample, project and then add the perturbation to the original image as shown in Algorithm 1 to get the perturbed image to conform to the input space of the model. To generate a successful adversarial perturbation, it is necessary and sufficient to have $v_t > 0$, as described in Section 3. We call our attack successful with $t$ queries to the model if the Bayesian optimization loop exits after $t$ iterations (line 12 in Algorithm 2), otherwise it is unsuccessful. Finally, we note that the final adversarial image can be obtained by upsampling the learned perturbation and adding to the original image as shown in Figure 1.

In this work, we focus on $\ell_\infty$-norm perturbations, where projection is defined as:

$$\Pi^\infty_{B(\mathbf{x_0},\epsilon)}(\mathbf{x}) = \min\left\{\max\{\mathbf{x_0} - \epsilon, \mathbf{x}\}, \mathbf{x_0} + \epsilon\right\}, \tag{3}$$

where $\epsilon$ is the given perturbation bound. The upsampling method can be linear or non-linear. In this work, we conduct experiments using nearest neighbor upsampling. A variational autoencoder (Kingma & Welling, 2014) or vanilla autoencoder could also be trained to map the low dimension perturbation to the original input space. We compare these different upsampling schemes in Section 5.3.1. The initial choice of the dataset $\mathcal{D}$ to form a prior can be done using standard normal distribution, uniform distribution or even in a deterministic manner (e.g. with Sobol sequences).

## 5 EXPERIMENTS

Our experiments focus on the untargeted attack setting where the goal is to perturb the original image originally classified correctly by the classification model to cause misclassification. We primarily consider performance of BAYES-ATTACK on ImageNet classifiers and compare its performance to other black-box attacks in terms of success rate over a given query budget. We also perform ablation studies on the MNIST dataset (Lecun et al., 1998) by examining different upsampling techniques and varying the latent dimension $d'$ of the optimization problem.

We define success rate as the ratio of the number of images successfully perturbed for a given query budget to the total number of input images. In all experiments, images that are already misclassified

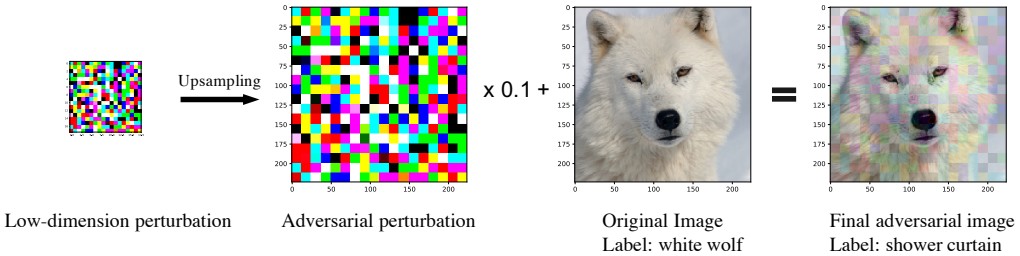

Figure 1: An illustration of a black-box adversarial attack performed by the proposed method BAYES-ATTACK on RESNET50 trained on ImageNet. Images from the left: first figure shows the learnt perturbation in low dimension $d' = 972(3 \times 18 \times 18)$; second figure is the final adversarial perturbation $(3 \times 224 \times 224)$ obtained by using nearest neighbor upsampling; third figure is the original image (note that the input size for RESNET50 is $3 \times 224 \times 224$) which is initially classified as *white/arctic wolf*; last image is the final adversarial image obtained by adding the adversarial perturbation to the original image. RESNET50 classifies the final adversarial image as *shower curtain* with high probability.

by the target network are excluded from the test set; only images that are initially classified with the correct label are attacked. For each method of attack and each target network, we compute the average and median number of queries used to attack among images that were successfully perturbed.

## 5.1 EMPIRICAL PROTOCOLS

We treat the latent dimension $d'$ used for running the Bayesian optimization loop as a hyperparameter. For MNIST, we tune the latent dimension $d'$ over $\{16, 64, 100, 256, 784\}$. Note that $784$ is the original input dimension for MNIST. While for ImageNet, we search for latent dimension $d'$ and shape over the range $\{48(3 \times 4 \times 4), 49(1 \times 7 \times 7), 100(1 \times 10 \times 10), 108(3 \times 6 \times 6), 400(1 \times 20 \times 20), 432(3 \times 12 \times 12), 576(1 \times 24 \times 24), 588(3 \times 14 \times 14), 961(1 \times 31 \times 31), 972(3 \times 18 \times 18)\}$. For ImageNet, the latent shapes with first dimension as 1 indicate that the same perturbation is added to all three channels while the ones with 3 indicate that the perturbation across channels are different. In case of ImageNet, we found that for ResNet50 and VGG16-bn different perturbation across channels work much better than adding the same perturbation across channels. While for Inception-v3, both seem to work equally well.

We initialize the GP with $n_0 = 5$ samples sampled from a standard normal distribution. For all the experiments in next section, we use expected improvement as the acquisition function. We also examined other acquisition functions (posterior mean, probability of improvement, upper confidence bound) and observed that our method works equally well with other acquisition functions. We independently tune the hyper-parameters on a small validation set and exclude it from our final test set. We used BoTorch[1] packages for implementation.

## 5.2 EXPERIMENTS ON IMAGENET

We compare the performance of the proposed method BAYES-ATTACK against NES (Ilyas et al., 2018), BANDITS-TD (Ilyas et al., 2019) and PARSIMONIOUS (Moon et al., 2019), which is the current state of the art among score-based black-box attacks within the $\ell_\infty$ threat model. On ImageNet, we attack the pretrained[2] ResNet50 (He et al., 2015), Inception-v3 (Szegedy et al., 2015) and VGG16-bn (Simonyan & Zisserman, 2014). We use 10,000 randomly selected images (normalized to [0, 1]) from the ImageNet validation set that were initially correctly classified.

We set the $\ell_\infty$ perturbation bound $\epsilon$ to 0.05 and evaluate the performance of all the methods for low query budgets. We use the implementation[3] and hyperparameters provided by Ilyas et al. (2019) for

---

[1]https://botorch.org/

[2]Pretrained models available at https://pytorch.org/docs/stable/torchvision/models

[3]https://github.com/MadryLab/blackbox-bandits

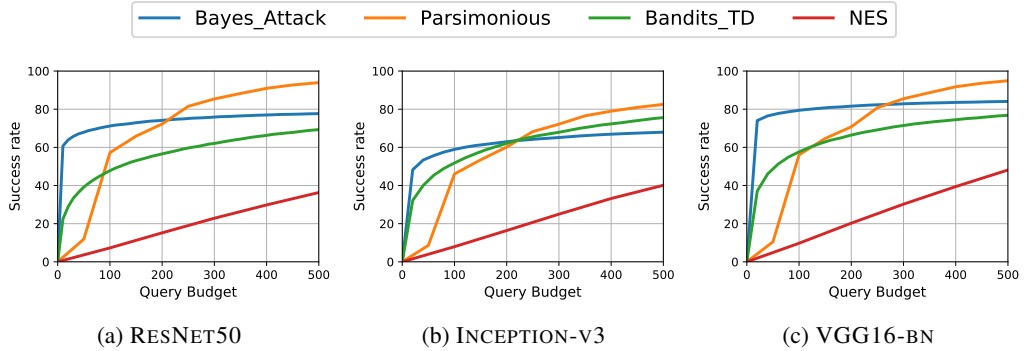

(a) RESNET50         (b) INCEPTION-V3         (c) VGG16-BN

Figure 2: Performance comparison for $\ell_\infty$ untargeted attacks on ImageNet classifiers. BAYES-ATTACK consistently performs better for low query budgets ($\leq 200$). Note that for NES, model queries are performed in batches of 100 as specified in Ilyas et al. (2018).

Table 1: Results for $\ell_\infty$ untargeted attacks on ImageNet classifiers with a query budget of 200

| CLASSIFIER | METHOD | SUCCESS RATE | AVERAGE QUERY | MEDIAN QUERY |
|---|---|---|---|---|
| RESNET50 | NES | 15.20% | 152 | 200 |
| | BANDITS-TD | 56.59% | 44 | 20 |
| | PARSIMONIOUS | 72.26% | 79 | 73 |
| | BAYES-ATTACK | **74.16%** | **17** | **6** |
| INCEPTION-V3 | NES | 20.27% | 152 | 200 |
| | BANDITS-TD | 66.44% | 40 | 16 |
| | PARSIMONIOUS | 70.75% | 80 | 73 |
| | BAYES-ATTACK | **81.60%** | **13** | **6** |
| VGG16-BN | NES | 16.41% | 152 | 200 |
| | BANDITS-TD | 62.10% | 45 | 20 |
| | PARSIMONIOUS | 60.16% | 84 | 74 |
| | BAYES-ATTACK | **62.95%** | **22** | **6** |

NES and BANDITS-TD. Similarly for PARSIMONIOUS, we use the implementation[4] and hyperparameters given by Moon et al. (2019).

Figure 2 compares the performance of the proposed method BAYES-ATTACK against the set of baseline methods in terms of success rate at different query budgets. We can see that BAYES-ATTACK consistently performs better than baseline methods for query budgets $< 200$. Even for query budgets $> 200$, BAYES-ATTACK achieves better success rates than BANDITS-TD and NES on ResNet50 and VGG16-bn. Finally, we note that for higher query budgets ($> 1000$), both PARSIMONIOUS and BANDITS-TD method perform better than BAYES-ATTACK.

To compare the success rate and average/median query, we select a point on the plots shown in Figure 2. Table 1 compares the performance of all the methods in terms of success rate, average and median query for a query budget of 200. We can see that BAYES-ATTACK achieves higher success rate with $80\%$ less average queries as compared to the next best PARSIMONIOUS method. Thus, we argue that although the Bayesian Optimization adversarial attack approach is to some extent a "standard" application of traditional Bayesian Optimization methods, the performance over the existing state of the art makes it a compelling approach particularly for the very low query setting. We also compare the average $\ell_2$ distortion of the generated adversarial perturbations in Appendix B.

---
[4]https://github.com/snu-mllab/parsimonious-blackbox-attack

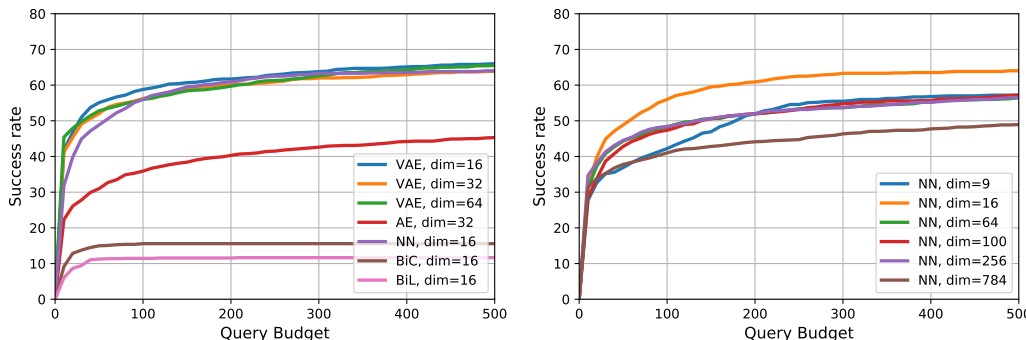

(a) Performance comparison with different upsampling schemes.

(b) Performance comparison with different latent dimension.

Figure 3: $\ell_\infty$ untargeted attacks on MNIST. *dim*: latent dimension used to run the Bayesian optimization, *NN*: Nearest neighbor interpolation, *BiC*: Bicubic interpolation, *BiL*: Bilinear interpolation, *AE*: autoencoder-based decoder, *VAE*: VAE-based generator (or decoder)

### 5.3  EXPERIMENTS ON MNIST

For MNIST, we use the pretrained network (used in Carlini & Wagner (2017)) with 4 convolutional layers, 2 max-pooling layers and 2 fully-connected layers which achieves $99.5\%$ accuracy on MNIST test set. We conduct $\ell_\infty$ untargeted adversarial attacks with perturbation bound $\epsilon = 0.2$ on a randomly sampled $1000$ images from the test set. All the experiments performed on MNIST follow the same protocols.

#### 5.3.1  UPSAMPLING METHODS

The proposed method requires an upsampling technique for mapping the perturbation learnt in the latent dimension to the original input dimension. In this section, we examine different linear and non-linear upsampling schemes and compare their performance on MNIST. The approaches we consider here can be divided into two broad groups: Encoder-Decoder based methods and Interpolation methods. For interpolation-based methods, we consider nearest-neighbor, bilinear and bicubic interpolation.

For encoder-decoder based approaches, we train a variational autoencoder (Kingma & Welling, 2014; Rezende et al., 2014) by maximizing a variational lower bound on the log marginal likelihood. We also consider a simple autoencoder trained by minimizing the mean squared loss between the generated image and the original image. For both the approaches, we run the Bayesian optimization loop in latent space and use the pretrained decoder (or generator) for mapping the latent vector into image space. For these approaches, rather than searching for adversarial perturbation $\delta$ in the latent space, we learn the adversarial image $\mathbf{x} + \delta$ directly using the Bayesian optimization.

Figure 3a compares the performance of different upsampling methods. We can see that Nearest Neighbor (*NN*) interpolation and VAE-based decoder perform better than rest of the upsampling schemes. However, the *NN* interpolation achieves similar performance to the VAE-based method but without the need of a large training dataset which is required for accurately training a VAE-based decoder.

#### 5.3.2  LATENT DIMENSION SENSITIVITY ANALYSIS

We perform a sensitivity analysis on the latent dimension hyperparameter $d'$ used for running the Bayesian optimization. We vary the latent dimension over the range $d' \in \{9, 16, 64, 100, 256, 784\}$. Figure 3b shows the performance of nearest neighbor interpolation method for different latent dimension. We observe that lower latent dimensions achieve better success rates than the original input dimension $d' = 784$ for MNIST. This could be because with increase in search dimension, Bayesian optimization needs more queries to find successful perturbation. We also note that for the case of latent dimension $d' = 9$, BAYES-ATTACK achieves lower success rates which could mean

that it is hard to find adversarial perturbations in such low dimension. We show the convergence plots of BAYES-ATTACK on ImageNet and MNIST in Appendix A.

## 6    CONCLUSIONS

We considered the problem of black-box adversarial attacks in settings involving constrained query budgets. We employed Bayesian optimization based method to construct a query efficient attack strategy. The proposed method searches for an adversarial perturbation in low dimensional latent space using Bayesian optimization and then maps the perturbation to the original input space using the nearest neighbor upsampling scheme. We successfully demonstrated the efficacy of our method in attacking multiple deep learning architectures for high dimensional inputs. Our work opens avenues regarding applying BO for adversarial attacks in high dimensional settings.

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

## A    CONVERGENCE OF BAYESIAN OPTIMIZATION

In this section, we show the convergence of Bayesian Optimization (BO) in terms of objective value versus the number of queries. Note that we have framed our objective of finding adversarial perturbation to be a maximization problem and we stop the iteration loop of BO once the objective value reaches a positive value. A positive objective value corresponds to a successful adversarial perturbation as described in Section 3.

Figure 4 shows the convergence of objective function in the BAYES-ATTACK on RESNET50 trained on ImageNet as described in Section 5.2. We run the BO in 972 dimensions $(3 \times 18 \times 18)$ and upsample the perturbation to the original input dimension of $150,528(3 \times 224 \times 224)$. The plot shows ten randomly chosen images from the validation set, with different colors representing different images.

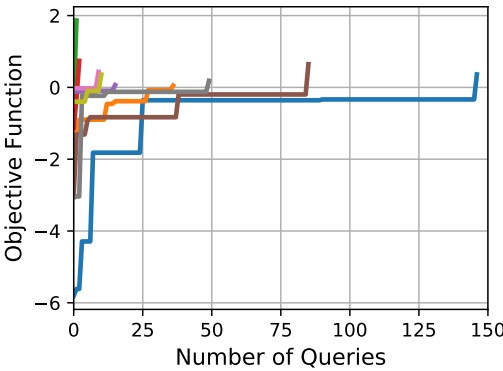

Figure 4: Convergence graphs on RESNET50 trained on ImageNet

We also show the convergence of the BO on MNIST by varying the latent dimension. Specifically, we compare the convergence with latent dimension $16(4 \times 4)$ and the original input dimension $784(28 \times 28)$. The plot is shown in Figure 5. Each color represents a test image, while dashed lines and solid lines represent runs of BO using 16 and 784 dimensions, respectively. As we can see from the graph, BO in 784 dimensions does not converge to a successful attack (i.e., objective value $> 0$) in $500$ iterations on either of the images, while BO with 16 dimensions on the same images finds the adversarial perturbation in less than 200 iterations. This aligns with our observation that with increase in latent dimension, it becomes harder for BO to find successful perturbation and indeed it would require much more queries than running BO in lower dimensions.

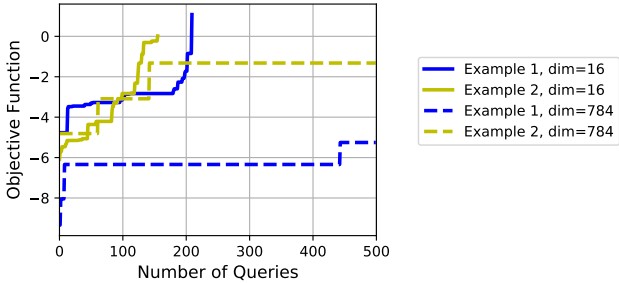

Figure 5: Convergence graphs on MNIST

## B    DISTORTION EFFICIENCY

We compare the average $\ell_2$ distortion per image of the proposed method BAYES-ATTACK with the current state-of-the-art methods including gradient-based approaches on RESNET50 trained on

ImageNet. We fix the query budget at 200 similar to the experiments described in Section 5.2 and compute the distortion using only the successful adversarial perturbations. As we can see from Table 2, the $\ell_2$ distortion of adversarial examples generated using BAYES-ATTACK is almost similar to the current state-of-the-art methods but achieves better attack success rate in low query budget regimes. Having said that, as in BANDITS-TD (Ilyas et al., 2019) and PARSIMONIOUS (Moon et al., 2019), our approach focuses on finding successful adversarial perturbations subject to a pre-defined maximum distortion specified in terms of $\ell_\infty$ distance.

Table 2: Average $\ell_2$ distortion of adversarial examples generated for $\ell_\infty$ untargeted attacks on RESNET50 trianed on ImageNet.

| METHOD | SUCCESS RATE | AVERAGE $\ell_2$ DISTORTION |
|---|---|---|
| BANDITS-TD | 56.59% | 18.65 |
| PARSIMONIOUS | 72.26% | 19.40 |
| BAYES-ATTACK | 74.16% | 19.14 |

