# OpenReview forum: "Black-box Adversarial Attacks with Bayesian Optimization"
_ICLR.cc/2020/Conference — Reject_

### Official Review · AnonReviewer2 · 2019-10-14
**Official Blind Review #2**

**Rating:** 3

**Review:**

The paper presents an idea on making adversarial attack on deep learning model. Since the space of input-output for the adversarial attach is huge, the paper proposes to use Bayesian optimization (BO) to sequentially select an attack.

Although the potential application of adversarial attack on deep learning model is interesting, the paper contribution and the novelty are limited giving the fact that there is another related paper published [1].

The authors in [1] consider using Bayesian optimization to make adversarial attack for model testing. In particular, they have considered the deep learning model. Then, they extend to multi-task settings. There is a big overlapping between the idea in [1] and the current paper.

The paper presentation and writing is high quality although the paper is a bit over-length.

[1] Gopakumar, Shivapratap, et al. "Algorithmic assurance: an active approach to algorithmic testing using Bayesian optimisation." Advances in Neural Information Processing Systems. 2018.



**Experience Assessment:**

I have published in this field for several years.

**Review Assessment: Checking Correctness Of Derivations And Theory:**

I carefully checked the derivations and theory.

**Review Assessment: Checking Correctness Of Experiments:**

I assessed the sensibility of the experiments.

**Review Assessment: Thoroughness In Paper Reading:**

I read the paper at least twice and used my best judgement in assessing the paper.

---

> ### Author Response · Authors · 2019-11-08
> **Response to reviewer #2**
>
> Thanks for pointing out the related work. We will add this to our final version. In regards to the comparison with [1] we respectfully disagree with the reviewer for the following reasons:
>
> 1) In [1], the authors limited themselves to a specific class of perturbations which only encompasses shearing and rotation distortion. Furthermore, the set of perturbations ensures that the space over which the optimization is performed is just 3 dimensions, which is an ideal BO use case as BO is particularly effective for lower dimensional optimization. However, in our case the perturbations belong to high dimensional spaces, i.e., 784 for MNIST, 224x224x3 for VGG-16bn and ResNet and finally 299x299x3 for Inception-v3. Given the huge difference in the dimensionality of the problem considered in our paper and [1], it is not clear how Bayesian optimization (BO) in its vanilla form can be by default extended to such high dimensional settings. It is particularly challenging to make BO work in high dimensional optimization problems in a constrained setting which is something we address in this paper, which [1] does not.
>
> 2) Moreover, the setting in our paper involves a constrained optimization problem, where the constraint is specified by an $\ell_{\infty}$ norm. Thus, the novelty of our algorithm with the dimensionality reduction part therein goes above and beyond the algorithm in [1]. As shown in Figure 3 in Section 5.3, straight forward application of BO to finding adversarial perturbation in the input space leads to sub-standard results. In this paper, we show how we can run BO in low dimensional space to generate successful adversarial perturbations bounded in $\ell_\infty$ norm in a query efficient manner.
>
> 3) We also feel that our work considers this problem in much more comprehensive detail than [1]: we perform experiments across several datasets and deep learning architectures, and compare various methods of upsampling to maximize attack success rate. Most importantly, the primary objective of our work is to present an attack method that performs well under a tightly constrained query budget. However, the setting in [1] is completely agnostic to query budgets. It is non-trivial to get an understanding of the algorithm in [1] in query constrained settings.
>
>
> In summary, we study the efficacy of black-box attacks in successfully generating adversarial perturbations for deep learning classifiers in severely query limited settings, e.g., with a query budget of 100-200 which [1] does not consider at all while several recent papers have considered query budgets on the order of 10,000 or 100,000. The techniques introduced in this paper to handle high dimensional optimization problems go above and beyond the techniques introduced in [1]. It is not clear from [1] as to how the algorithm performs in high dimensional frameworks like ours in a constrained setting.
>
> Reference:
> [1] Gopakumar, Shivapratap, et al. "Algorithmic assurance: an active approach to algorithmic testing using Bayesian optimisation." Advances in Neural Information Processing Systems. 2018.

---

### Official Review · AnonReviewer1 · 2019-10-24
**Official Blind Review #1**

**Rating:** 6

**Review:**

This paper proposed a BO-based black-box attack generation method. In general, it is very well written and easy to follow. The main contribution is to combine BO with dimension reduction, which leads to the effectiveness in generating black-box adversarial examples in the regime of limited queries.  However, I still have some concerns about this paper.

1) The benefits of BO? It seems that the step of dimension reduction is crucial to make BO scablable to high-dimensional problems. I wonder if the gradient estimation-based attack methods can apply the similar trick and yield the similar performance. That is, one can solve problem min_{\delta} attack_loss( x, y, g(\delta) ) by estimating gradients via finite-difference of function values, where g(\cdot) is the dimension-reduction operator, and \delta is the low-dimensional perturbation. Such a baseline is not clear in the paper, and the comparison with (Tu et al., 2019) is not provided in the paper.

2) Moreover, in experiments, it seems that only query-efficiency was reported. What about distortion-efficiency for BO-based attack? For $\ell_\infty$-attacks, the other $\ell_p$ norms can be used as distortion metrics. I wonder what perturbation does the BO method converge to. It was shown in (https://arxiv.org/pdf/1907.11684.pdf, Table 1) that BO usually leads to larger \ell_1 and \ell_2 distortion.

3) It might be useful to show the convergence of BO in terms of objective value versus iterations/queries. This may give a clearer picture on how BO works in the attack generation setting.

4) Minor comment: In related work "Bayesian optimization has played a supporting role in several methods,
including Tu et al. (2019), where ...." However,  Tu et al. (2019) does not seem using BO and ADMM.

############ Post-feedback ##########
Thanks for the clarification and the additional experiments. I am satisfied with the response, and have increased my score to 6.


**Experience Assessment:**

I have published in this field for several years.

**Review Assessment: Checking Correctness Of Derivations And Theory:**

I assessed the sensibility of the derivations and theory.

**Review Assessment: Checking Correctness Of Experiments:**

I assessed the sensibility of the experiments.

**Review Assessment: Thoroughness In Paper Reading:**

I read the paper at least twice and used my best judgement in assessing the paper.

---

> ### Author Response · Authors · 2019-11-08
> **Reply to reviewer #1**
>
> Thank you for the insightful comments. We address the issues below:
>
> 1) The benefits of BO? ...
> A: This idea of finding adversarial perturbation in low dimension has already been applied in gradient estimation based attack methods.  [1] used data-dependent priors — a tiling based approach, to improve the black-box adversarial attack performance. We compare this method with Bayes-Attack in section 5.2 and show that the proposed method consistently performs better than [1] for low query budgets on standard ImageNet models.
>
> In this paper, we have focused on untargeted $\ell_\infty$ attacks and compared the baseline models in that domain. Although we do not compare directly to [3] as they perform $\ell_2$ attacks, we compare the autoencoder-based dimension reduction technique introduced in [3] in section 5.3.  Additionally, in Appendix G of [1], authors compared with [3] and showed that [1] achieves significantly better performance.
>
> 2) Moreover, in experiments, it seems that only query-efficiency was reported. What about distortion-efficiency...
> A: Below, we compare the average $\ell_2$ distortion per image of our method Bayes-Attack with the current state of the art methods including gradient-based approach Bandits-TD [1] on ResNet50 trained on ImageNet. We fixed the query budget at 200 and computed the $\ell_2$ distortion using only successful adversarial perturbations.
>
> Method      Avg $\ell_2$-distortion  Success Rate
> Bandits-TD [1]            18.65                56.59%
> Parsimonious [2]        19.40                72.26%
> Bayes-Attack               19.14                 74.16%
>
> As we can see from the table, the $\ell_2$ distortion of adversarial examples generated using Bayes-Attack is almost similar to the gradient-based method [1] but achieves better attack success rate. Having said that, as in [1] and [2],  our approach focuses on finding successful adversarial perturbations subject to a pre-defined maximum distortion specified in terms of $\ell_p$ distance.
>
> 3) It might be useful to show the convergence of BO in terms of objective value versus iterations/queries. This may give a clearer picture on how BO works in the attack generation setting.
> A: https://i.postimg.cc/9XJ8tfqH/bayes-opt-iter.png
> Note that we have framed our objective of finding adversarial perturbation to be a maximization problem and we stop the iteration loop of BO once the objective value reaches a positive value. A positive objective value corresponds to a successful adversarial perturbation as described in Section 3. We will add this to our final version.
>
> 4) Minor comment...
> A: We have updated the reference. Thanks for pointing this out.
>
> References:
> [1]  Andrew Ilyas, Logan Engstrom, and Aleksander Madry. Prior convictions: Black-box adversarial attacks with bandits and priors. In International Conference on Learning Representations, 2019.
> [2] Seungyong Moon, Gaon An, and Hyun Oh Song. Parsimonious black-box adversarial attacks via efficient combinatorial optimization. Proceedings of the 36th International Conference on Machine Learning, ICML 2019.
> [3] Chun-Chen Tu, Pai-Shun Ting, Pin-Yu Chen, Sijia Liu, Huan Zhang, Jinfeng Yi, Cho-Jui Hsieh, and Shin-Ming Cheng. Autozoom: Autoencoder-based zeroth-order optimization method for attacking black-box neural networks. In The Thirty-Third AAAI Conference on Artificial Intelligence, AAAI 2019.

---

> > ### Comment · AnonReviewer1 · 2019-11-08
> > **Thanks for clarification**
> >
> > Thanks for the prompt response. My concerns have largely been alleviated.
> >
> > In the image of response 3, what does the color refer to? Also how is the convergence of BO sensitive to the dimension of latent space for attacks under MNIST and ImageNet?
> >
> > In the paper, the author mentioned "We observe that lower latent dimensions achieve better success rates than the original input dimension d0 = 784 for MNIST. This could be because with increase in search dimension, Bayesian optimization needs more queries to find successful perturbation." This also means the possible poor convergence of BO as the dimension increases (dimension-dependent effect in convergence). Do we expect to see this from the newly added convergence results?
> >
> > I am glad to increase my score once the revision has a better clarification on the pros and cons of BO.

---

> > > ### Author Response · Authors · 2019-11-13
> > > **Convergence Plots**
> > >
> > > Plots shown here (https://i.postimg.cc/zXZJJhTc/resnet50-bayes-opt-iter.png) correspond to the objective function over time in the Bayes-Attack on ResNet50 trained on ImageNet described in Section 5.2. We run the BO in 972 dimensions (3x18x18) and upsample the perturbation to the original input dimension of 150,528 (3x224x224). The plot shows ten randomly chosen images from the experiment, with different colors representing different images.
> > >
> > > We also show the convergence of the BO on MNIST by varying the latent dimension. Specifically, we compare the convergence with latent dimension 16 and original input dimension 784. The plot can be seen here: https://i.postimg.cc/9FMD7ZDF/mnist-bayes-opt-iter.png. Each color represents a test image, while dashed lines and solid lines represent runs of BO using 16 and 784 dimensions, respectively. As we can see from the graph, BO in 784 dimensions does not converge to a successful attack (i.e., objective value > 0) in 500 iterations on either of the images, while BO with 16 dimensions on the same images finds the adversarial perturbation in less than 200 iterations. This aligns with our observation that with increase in latent dimension, it becomes harder for BO to find successful perturbation and indeed it would require much more queries than running BO in lower dimensions.

---

### Official Review · AnonReviewer3 · 2019-10-24
**Official Blind Review #3**

**Rating:** 1

**Review:**

This paper applies Bayesian optimisation (BO), a sample efficient global optimisation technique, to the problem of finding adversarial perturbation. First, the application is starightforward application of BO to the well-known problem of adversarial perturbation. Nothing innovative. Second, the paper addresses the high dimensional optimisation with simple upsampling technique like nearest-neighbour, without even trying their hands dirty by using one of many many high-dimensional Bayesian optimisation algorithm (a quick Google search will reveal them), The work  thus fail in thoroughness also. Third, adversarial perturbation are known to exist even around the image such that even a simple gradient descet optimisation starting from the target image would be able to provide perceptually small perturbation (it does not have to the smallest to be perceptually small). Hence, the impact is also missing. Thus accroding to me this paper is not good enough for acceptance.

**Experience Assessment:**

I have published in this field for several years.

**Review Assessment: Checking Correctness Of Derivations And Theory:**

I assessed the sensibility of the derivations and theory.

**Review Assessment: Checking Correctness Of Experiments:**

I assessed the sensibility of the experiments.

**Review Assessment: Thoroughness In Paper Reading:**

I read the paper at least twice and used my best judgement in assessing the paper.

---

> ### Author Response · Authors · 2019-11-08
> **Response to reviewer #3**
>
> We thank the reviewer for his comments. We address the concerns below:
>
> R:  First, the application is a straight forward application of BO to the well-known problem of adversarial perturbation...
> A: We agree with the reviewer that this paper is a direct application of BO to the problem of black-box adversarial attacks. However, most existing black-box adversarial attacks take into account per image query budgets of 10000 or sometimes even higher so as to generate successful attacks. The query constrained setting is something which has been largely understudied which brings the question whether successful attacks can be designed for potentially high dimensional models/images such as ImageNet with query budgets of 200 or even lower. While zeroth order optimization algorithms based on finite difference approximation have been used a lot for handling such problems, their performance has been found to be ineffective in constrained query budget setting. Moreover the query efficiency of BO as compared to zeroth order counterparts makes it an ideal candidate for such a setting. In the face of high dimensional constrained optimization, with a constrained query budget, we propose a simple and computationally efficient method to come up with successful adversarial attacks.
>
> R:  Second, the paper addresses the high dimensional optimisation with simple upsampling technique like nearest-neighbour...
> A: We agree with the reviewer that there have been recent works as far as handling high dimensional Bayesian optimization problems are concerned. However, in our search, we found the highest dimension of the optimization problem handled by the aforementioned setups in experiments to be 150 [1]. It is worth noting that the dimensionality of the optimization problem we handle in our setup ranges from 784 for MNIST to 270000 for ImageNet. So, it is not clear how the previous algorithms handling high dimensional optimization algorithms with BO scale to the dimensions we are considering in this paper. We would like to ask the reviewer to point us to any specific references in this regard which considers employing BO for high-dimensional optimization problems with dimension in the 100000 range. The upsampling technique used in our method is simple as rightly pointed out by the reviewer, but at the same time it is very effective and outperforms much more complex black-box attacks at low query budgets, as we demonstrate across multiple datasets on multiple architectures.
>
> R:  Third, adversarial perturbation are known to exist even around the image such that even a simple gradient descent optimisation...
> A: In this work, we focus on the black-box setting where the model weights, architecture, and gradient information is not available to the attacker. Gradient descent in itself is a first order optimization method requiring gradient, i.e., first order information is not applicable to the black-box setup. In this setting, information about the network can be obtained only through querying the target network in order to access loss function values. There are several methods [2, 3] in this setting that estimate gradients using finite difference approximations based on the model outputs and then subsequently use the estimated gradients for finding adversarial perturbations. As shown in section 5.2, the proposed method consistently performs better than the current state-of-the-art methods including gradient-based methods for low-query budgets across several deep learning architectures on ImageNet.
>
> References:
> [1] K.Kandasamy, J. Schneider and B. Poczos, High Dimensional Bayesian Optimisation and Bandits via Additive Models, International Conference on Machine Learning, 2015.
> [2] Andrew Ilyas, Logan Engstrom, and Aleksander Madry. Prior convictions: Black-box adversarial attacks with bandits and priors. In International Conference on Learning Representations, 2019.
> [3] Andrew Ilyas, Logan Engstrom, Anish Athalye, and Jessy Lin. Black-box adversarial attacks with limited queries and information. In Proceedings of the 35th International Conference on Machine Learning, ICML 2018.

---

### Author Response · Authors · 2019-11-15
**Revision**

We would like to thank all the reviewers for their reviews and insightful comments. We have uploaded a revised draft.

- We have added a comparison of the distortion efficiency of our method in terms of $\ell_2$ distance with the current state-of-the-art methods including gradient based methods.
- We have also added plots to show the convergence of our method with the number of queries on ImageNet and MNIST. We have also compared the convergence of Bayesian Optimization by varying the latent dimension on MNIST.
- We have added the reference suggested by Reviewer #2 and discussion pertaining to it in the related work.

---

### Decision · Program_Chairs · 2019-12-19

**Decision:**

Reject

**Comment:**

The paper proposes a Bayesian optimization approach to creating adversarial examples. The general idea has been in the air for some years, and over the last year especially there have been a number of approaches using BayesOpt for this purpose. Reviewers raised concerns about differences between this approach and related work, and practical challenges in general for using BayesOpt in this domain (regarding dimensionality, etc.). The authors provided thoughtful responses, although some of these concerns still remain. The authors are encouraged to address all comments carefully in future revisions, which a sufficiently substantial that the paper would benefit from additional review.